# DSLR: Diversity Enhancement and Structure Learning for Rehearsal-based Graph Continual Learning

## ABSTRACT

We investigate the replay buffer in rehearsal-based approaches for graph continual learning (GCL) methods. Existing rehearsal-based GCL methods select the most representative nodes for each class and store them in a replay buffer for later use in training subsequent tasks. However, we discovered that considering only the class representativeness of each replayed node makes the replayed nodes to be concentrated around the center of each class, incurring a potential risk of overfitting to nodes residing in those regions, which aggravates catastrophic forgetting. Moreover, as the rehearsal-based approach heavily relies on a few replayed nodes to retain knowledge obtained from previous tasks, involving the replayed nodes that have irrelevant neighbors in the model training may have a significant detrimental impact on model performance. In this paper, we propose a GCL model named **D**iversity enhancement and **S**tructure **L**earning for **R**ehearsal-based graph continual learning (DSLR). Specifically, we devise a coverage-based diversity (CD) approach to consider both the class representativeness and the diversity within each class of the replayed nodes. Moreover, we adopt graph structure learning (GSL) to ensure that the replayed nodes are connected to truly informative neighbors. Extensive experimental results demonstrate the effectiveness and efficiency of DSLR. Our source code is available at https://anonymous.4open.science/r/DSLR-F525.

## 1 INTRODUCTION

Training a new model for every large stream of new data is not only time-consuming but also cost-intensive. Hence, continual learning [17, 19, 20, 25, 26, 31, 32, 34] has become crucial in addressing this challenge. Continual learning emphasizes efficient learning from newly introduced data without retraining the model on the entire dataset, enabling the preservation of previously acquired knowledge. However, there is a risk of forgetting the previously acquired knowledge when the model is trained on new data, resulting in the performance decline. This phenomenon is referred to as catastrophic forgetting, and the ultimate goal of continual learning models is to mitigate catastrophic forgetting.

Continual learning approaches are broadly categorized into the regularization-based approach [10, 12, 15, 26, 30], architectural approach [32], and rehearsal-based approach [17, 23, 26, 27, 34]. The rehearsal-based approach, which will be investigated in this paper, involves storing a small amount of data from the current task for later use when training subsequent tasks. This collection of data

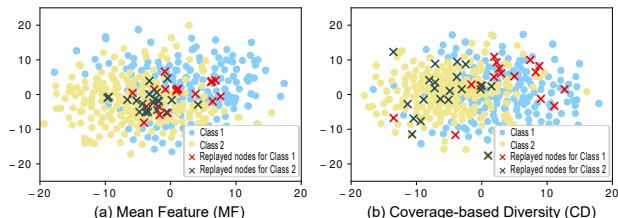

**Figure 1: T-SNE visualization of node embeddings belonging to class 1 and 2 along with the replayed nodes for each class selected using (a) MF and (b) CD in Citeseer dataset.**

is referred to as the replay buffer. The rehearsal-based approach is widely recognized as the most effective for addressing continual learning among other approaches for continual learning [2].

One of the state-of-the-art rehearsal-based approaches for graph continual learning (GCL), named ER-GNN [34], selects the most representative nodes for each class and stores them in a replay buffer for later use in training subsequent tasks. Specifically, ER-GNN proposed the mean feature (MF) approach, which computes the average of the node features in each class, and store nodes that are close to the average of each class in the replay buffer. However, while this approach considers the class representativeness of each replayed node, we argue that the diversity of the replayed nodes within each class is overlooked. Fig. 1 (a) shows nodes belonging to two different classes in Citeseer dataset, along with the replayed nodes for each class obtained using MF. We observe that the replayed nodes for each class are mainly concentrated around the center of each class in the embedding space, incurring a potential risk of overfitting to nodes residing in those regions, which aggravates catastrophic forgetting. As a simple remedy for considering both the class representativeness and the diversity within each class of the replayed nodes, we devise a coverage-based diversity (CD) approach for selecting nodes to be replayed. By doing so, the replayed nodes can be evenly distributed across the embedding space of each class (See Fig. 1 (b)), indicating that overfitting to specific regions can be alleviated.

However, we observed that emphasizing the diversity aspect using CD can lead to another issue: some replayed nodes (i.e., those near the decision boundary as shown in Fig. 1 (b)) would be inevitably connected to many nodes from different classes. Concretely, Table 1 shows that using CD instead of MF leads to a lower homophily ratio[1] of the replayed nodes for each class. This in turn results in a low homophily ratio of the replayed nodes, which is harmful for training Graph Neural Networks (GNNs) whose performance is known to degrade on graphs with a low homophily ratio [5, 33] (See Fig. 2 where the replayed nodes with a lower homophily incurs

---

[1]Homophily ratio is a measure that indicates the proportion of neighbors connected to a specific node belonging to the same class.

**Table 1: Homophily ratio of the replayed nodes using MF & CD in Citeseer dataset.**

|  | MF | CD |
|---|---|---|
| Class 1 | 0.68 ± 0.43 | 0.57 ± 0.45 |
| Class 2 | 0.91 ± 0.24 | 0.92 ± 0.22 |
| Class 3 | 0.82 ± 0.28 | 0.76 ± 0.40 |
| Class 4 | 0.88 ± 0.26 | 0.82 ± 0.36 |

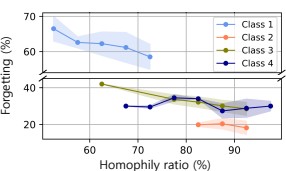

**Figure 2: Forgetting over various homophily ratios of the replayed nodes in Citeseer dataset.**

the model to more severely forget[2] previously acquired knowledge[3]). On the other hand, it is important to note that the increase in the forgetting performance begins to slow down beyond a certain point, and the variance begins to increase. This indicates that simply increasing the homophily ratio of the replay buffer deteriorates the model robustness, and thus is not an effective remedy.

In the light of these issues regarding the replay buffer of rehearsal-based approaches for GCL, we propose a GCL model named Diversity enhancement and Structure Learning for Rehearsal-based graph continual learning (DSLR), that consists of two major components: **Component 1** – Selecting the replayed nodes aiming at considering both the class representativeness and the diversity within each class, and **Component 2** – Reformulating the graph structure aiming encouraging the replayed nodes to be connected to *truly informative neighbors*. Note that we define truly informative neighbors as those that not only belong to the same class with the target node (i.e., homophily), but also share similar structure (i.e., structural proximity).

More precisely, we devise a coverage-based diversity (CD) approach for selecting nodes to be replayed (Component 1). The main idea of CD is to select diverse nodes while considering the class representativeness, ensuring that the replayed nodes effectively represent the entire embedding space of their own classes, which helps alleviate catastrophic forgetting. On the other hand, as the rehearsal-based approach heavily relies on a few replayed nodes to retain knowledge obtained from previous tasks, involving the replayed nodes that have irrelevant neighbors in the model training may have a significant detrimental impact on model performance. To address this issue, we adopt graph structure learning (GSL) [3, 33, 35] to reformulate the graph structure in a way that allows the replayed nodes to be connected to truly informative neighbors, so that high-quality information can be propagated into the replayed nodes through message passing (Component 2). The main idea of GSL is to add informative neighbors to the replayed nodes, and selectively delete edges between replayed nodes and their neighbors, both of which contribute to enhancing the effectiveness of the replay buffer. More precisely, we train a link predictor using both the class labels and the graph structure to capture the homophily and structural proximity, respectively, which helps us discover truly informative neighbors of the replayed nodes. Last but not least, we devise an efficient candidate selection method to improve the scalability of the graph structure learning.

Our contributions are summarized as follows:

[2]Forgetting in continual learning refers to the extent to which the accuracy of a particular task decreases after completing all tasks, e.g., if the accuracy of Task 1 was 90% and dropped to 50% after completing all tasks, the forgetting of Task 1 is 40%.
[3]Please refer to Appendix A for the detailed experimental setup.

(1) We emphasize the consideration of diversity when selecting the replayed nodes, a factor that has been overlooked in existing methods, to ensure that replayed nodes can effectively represent the entire data of their respective class.
(2) We present a novel discovery that emphasizes the substantial influence of the quality of neighbors surrounding the replayed nodes on the overall model performance. To this end, we adopt the graph structure learning to ensure that the replayed nodes are connected to truly informative neighbors.
(3) Extensive experiments show that DSLR outperforms state-of-the-art GCL methods, even with a small replay buffer size.

## 2 RELATED WORKS

We provide a concise overview of related work in this section. A complete discussion is in Appendix B.

***Graph Neural Networks.*** GNNs aggregate information from neighboring nodes, enabling them to capture both the structure and features of a graph, including more intricate graph patterns. One popular GNN model is Graph Convolutional Networks (GCN) [11], which introduces semi-supervised learning on graph-structured data using a convolutional neural network. GCN employs spectral graph convolution to update node representations based on their neighbors. Another approach, Graph Attention Networks (GAT) [24], employs attention mechanisms to assign varying weights to neighbors based on their importance.

***Graph Structure Learning.*** Real-world graphs contain incomplete structure. To alleviate the effect of noise, recent studies have focused on enriching the structure of the graph. The objective of these studies is to mitigate the noise in the graph and improve the performance of graph representation learning by utilizing refined data. GAUG [33] leverages edge predictors to effectively encode class-homophilic structure, thereby enhancing intra-class edges and suppressing inter-class edges within a given graph structure. IDGL [4] jointly and iteratively learns the graph structure and embeddings optimized for enhancing the performance of the downstream task. The application of structure learning in these methods improves the performance of downstream tasks by refining the incomplete or noisy structure of the graph, taking into consideration the inherent characteristics of real-world graphs.

***Continual Learning.*** Continual learning is a methodology in which a model learns from a continuous stream of datasets while retaining knowledge from previous tasks. However, as the model progresses through tasks, it often experiences a decline in performance due to forgetting the knowledge acquired from past tasks. This phenomenon is referred to as catastrophic forgetting. The primary objective of continual learning is to minimize catastrophic forgetting, and there are three main approaches employed in continual learning methods: 1) Rehearsal-based approach aims to select and store data from previous tasks for later use in training subsequent tasks. In the context of GCL, the selected nodes are referred to as replayed nodes, and the set of the replayed nodes is called replay buffer. The primary objective of the rehearsal-based approach is to carefully select the optimal replayed nodes that prevent the model from forgetting knowledge acquired in previous tasks. 2) Architectural approach involves modifying the model's architecture based on the task. If the model's capacity is deemed insufficient to

effectively learn new knowledge, the architecture is expanded to accommodate the additional requirements. 3) Regularization-based approach aims to regularize the model's parameters in order to minimize catastrophic forgetting while learning new tasks. This approach focuses on preserving parameters that were crucial for learning previous tasks, while allowing the remaining parameters to adapt and learn new knowledge.

# 3 PRELIMINARIES

**Graph Neural Networks.** A graph is represented as $\mathcal{G} = (A, X)$, where $A \in \mathbb{R}^{n \times n}$ is the adjacency matrix and $X \in \mathbb{R}^{|V| \times d}$ is the feature matrix with $d$ as the number of features. $V = \{v_i\}_{i=1}^n$ is the set of nodes, and $|V| = n$. In this paper, we use GAT [24] as the backbone whose aggregation scheme is defined as: $h_i^{(l)} = \sigma(\sum_{j \in \mathcal{N}(i)} \alpha_{ij} W^{(l)} h_j^{(l-1)})$, where $\alpha_{ij}$ is the attention score for the first-order neighbors of node $v_i$, $h_i^l$ is the $l$-th layer representation of node $v_i$, $\mathcal{N}(i)$ is the neighboring nodes of node $v_i$, $\sigma$ is an activation function, and $W^{(l)}$ is a trainable weight matrix at the $l$-th layer. We use the cross-entropy loss for the node classification task defined as follows:

$$\mathcal{L}_{\mathcal{D}}(\theta; A, X) = \frac{1}{|\mathcal{D}|} \sum_{v_i \in \mathcal{D}} l(\text{softmax}(h_i^L), y_i). \quad (1)$$

where $\mathcal{D}$ is the set of labeled nodes, $h_i^L$ is the final representation of node $v_i$[4], and $y_i$ is the label of node $v_i$.

**Continual Learning Setting.** We use $\mathcal{T} = \{T_1, T_2, \cdots, T_M\}$ to denote the set of tasks, where $M$ is the number of tasks. Then, the set of graphs in $M$ tasks is defined as follows:

$$\mathcal{G} = \{\mathcal{G}^1, \mathcal{G}^2, ..., \mathcal{G}^M\}, \text{ where } \mathcal{G}^t = \mathcal{G}^{t-1} + \Delta\mathcal{G}^t, \quad (2)$$

where $\mathcal{G}^t = (A^t, X^t)$ is the attributed graph given in Task $T_i$, and $\Delta\mathcal{G}^t = (\Delta A^t, \Delta X^t)$ is the change of the node attributes and the graph structure in Task $T_t$. The purpose of graph continual learning is to train models ($\text{GNN}_{\theta^1}, \text{GNN}_{\theta^2}, ..., \text{GNN}_{\theta^M}$) from streaming data where $\theta^t$ is the parameter of the GNN model in Task $T_t$. In this work, we focus on the node classification task under the class-incremental setting [6], where unseen classes are continually introduced as the task progresses.

# 4 PROPOSED METHOD

In this section, we describe our proposed method, DSLR, in detail. Fig. 3 presents the overall architecture of DSLR.

## 4.1 Coverage-based Diversity for Replay Buffer

Recall that in Fig. 1 (a) we observed that the replayed nodes of an existing replay technique, i.e., mean feature (MF), are concentrated around the center of each class in the embedding space, incurring a potential risk of overfitting to nodes residing in those regions, which aggravates catastrophic forgetting. To address this issue, we propose a coverage-based diversity (CD) approach that considers both the class representativeness and the diversity within each class of the replayed nodes. The main idea of CD is to utilize distance-based criteria to define the coverage for each node.

---
[4]For simplicity, we denote $h_i^L$ as $h_i$ from here.

**Buffer Selection**. The coverage of node $v_i$, i.e., $C(v_i)$, refers to the set of nodes belonging to the same class as node $v_i$ while being located within a certain distance in the embedding space from node $v_i$. Given node $v_i$ and its embedding $h_i = \text{GNN}(v_i)$, the coverage of node $v_i$ is formally defined as follows:

$$C(v_i) = \{v_j \mid dist(h_i, h_j) < d, \ y_i = y_j\}, \text{ where } d = r \cdot E(v_i), \quad (3)$$

where $dist(h_i, h_j)$ is the Euclidean distance between the embedding of nodes $v_i$ and $v_j$, $E(v_i)$ denotes the average of the sum of the pairwise distances between the embedding of training nodes that belong to the same class with node $v_i$, and $r$ is a hyperparameter.

The objective of CD is to select $e_l$ nodes to be replayed in a manner that maximizes the number of nodes included in the union of their coverage as follows:

$$\mathcal{B}_{C_l} = \text{argmax}_{\{v_{b_1}, \cdots, v_{b_{e_l}} \mid v_{b_1}, \cdots, v_{b_{e_l}} \in train_{C_l}\}} \left| Cover(\{v_{b_1}, \cdots, v_{b_{e_l}}\}) \right|, \quad (4)$$

$$\text{where } Cover(\{v_1, \cdots, v_n\}) = C(v_1) \cup \cdots \cup C(v_n). \quad (5)$$

where $\mathcal{B}_{C_l}$ is the set of replayed nodes of class $C_l$, where $|\mathcal{B}_{C_l}| = e_l$, $train_{C_l}$ is the training nodes of class $C_l$, and $\left| Cover(\{v_{b_1}, \cdots, v_{b_{e_l}}\}) \right|$ refers to the cardinality of $Cover(\{v_{b_1}, \cdots, v_{b_{e_l}}\})$. We employ a greedy algorithm to solve Equation 4, given its NP-hard complexity. The detailed algorithm is described in Appendix H, and Table 5 illustrates its competitive training speed through scalability analysis. Through CD, we select $e_l$ replayed nodes for each class $C_l$ and store them in the replay buffer, $\mathcal{B}$, for training subsequent tasks. $e_l$ is assigned proportionally to the number of nodes belonging to each class in the training set, i.e., for $e_l$ of class $C_l$ at task $T_t$, we set $e_l = \frac{|train_{C_l}|}{\sum_{t=1}^{t-1} \sum_{C_k \in \mathbb{C}_{T_t}} |train_{C_k}|} \cdot |\mathcal{B}|$, where $\mathbb{C}_{T_t}$ denotes the set of classes in task $T_t$. It is important to note that by ensuring that the coverage covers the maximum number of nodes, we not only consider the class **representativeness**, but also the **diversity** within each class of the replayed nodes. Eventually, we expect the selected $e_l$ replayed nodes through the aforementioned process to comprehensively represent the entire data distribution of class $C_l$. In Fig. 1 (b), we show that the nodes selected based on CD are indeed evenly distributed across the embedding space, preventing overfitting to specific regions, which helps alleviate catastrophic forgetting.

## 4.2 Structure Learning for Replay Buffer

Now that we have selected the replayed nodes by using CD, it is crucial to ensure that the replayed nodes are connected to informative neighbors so that high-quality information can be aggregated. This is especially important as the rehearsal-based approach heavily relies on a few replayed nodes, implying that involving the replayed nodes that have irrelevant neighbors in the model training may have a significant detrimental impact on model performance. To address this issue, we adopt graph structure learning (GSL) [35] to reformulate the graph structure in a way that allows the replayed nodes to be connected to truly informative neighbors, so that high-quality information can be propagated into the replayed nodes through message passing. More precisely, we train a link prediction module (Section 4.2.1), and use it to compute the link prediction score for each edge between a replayed node and candidates to be connected. Based on the scores, we decide whether to add or delete the edges (Section 4.2.2).

Figure 3: Overall architecture of DSLR. Upper boxes illustrate the comprehensive process of GCL using DSLR. After node classification, replayed nodes are selected (lower left box) and their structure is refined as new nodes are introduced (lower right box). The refined graph is then utilized for subsequent downstream tasks.

4.2.1 **Training Link Prediction Module**. Given the a graph in each task $T_t$, we train a GNN-based link prediction module. The main goal of the link prediction module is to discover *truly informative neighbors* of the replayed nodes, i.e., those that not only belong to the same class (i.e., homophily), but also share similar structure (i.e., structural proximity), so that they can be later used to refine the graph structure. To this end, we employ two loss functions to train the link prediction module, i.e., 1) link prediction loss and 2) node classification loss. The link prediction loss, which aims to capture the structural proximity, trains the link prediction module that predicts whether an edge exists between two given nodes. Specifically, we use a GNN-based link predictor, $\text{LP}_{\phi^t}$, to obtain a node embedding $z_i$ for each node $v_i$, and compute the cosine similarity, i.e., $Sim_{ij} = \frac{z_i \cdot z_j}{\|z_i\| \cdot \|z_j\|}$, and use it to compute the link prediction score $S_{ij}$ between node $v_i$ and node $v_j$, i.e., $S_{ij} = \frac{Sim_{ij}+1}{2}$. For task $T_t$, we construct a training set $\mathcal{D}_t^{link}$ that contain positive and negative edges sampled at each epoch to compute the link prediction loss $\mathcal{L}_{link}$ as follows:

$$\mathcal{L}_{link} = -\Big( \sum_{e_{ij} \in \mathcal{D}_t^{link}} (A_{ij}^t \log(S_{ij}) + (1 - A_{ij}^t)\log(1 - S_{ij})) \Big). \quad (6)$$

In addition to the unsupervised link prediction loss in Equation 6, we also include the node classification loss, which aims to capture the homophily aspect. It utilizes the class label information of training nodes, and is defined as follows:

$$\mathcal{L}_{node} = \beta \mathcal{L}_{\mathcal{D}_t^{tr}}(\theta^t; A^t, X^t) + (1 - \beta) \mathcal{L}_{\mathcal{B}}(\theta^t; A^t, X^t), \quad (7)$$

where $\mathcal{L}_{\mathcal{D}_t^{tr}}$ and $\mathcal{L}_{\mathcal{B}}$ denote the cross-entropy loss defined in Equation 1 for $\mathcal{D}_t^{tr}$ and $\mathcal{B}$, respectively, and $\mathcal{D}_t^{tr}$ and $\mathcal{B}$ are the set of the training nodes in the current task $T_t$ and the replay buffer $\mathcal{B}$ stored until task $T_{t-1}$, respectively. It is important to note that adding the node classification loss yields a higher homophily ratio of the replayed nodes than when only the link prediction loss is considered as will be demonstrated in Figure 9 in our experiments.

The final loss function for the link prediction module, $\mathcal{L}_{LP}$, with hyperparameter $\lambda$ to balance two losses is defined as follows:

$$\mathcal{L}_{LP} = \lambda \mathcal{L}_{link} + (1 - \lambda)\mathcal{L}_{node}. \quad (8)$$

In summary, the link prediction module aims to add/delete nodes with similar/dissimilar embeddings obtained by a GNN-based encoder, implying that it considers not only the class information (i.e., homophily) through $\mathcal{L}_{node}$, but also the graph structural information (i.e., structural proximity) through $\mathcal{L}_{link}$, which helps us discover truly informative neighbors of the replayed nodes.

4.2.2 **Structure Inference**. Having trained the link prediction module, we conduct structure inference to refine the structure of the replayed nodes using the trained link prediction module. The inference stage is divided into two phases, i.e., 1) edge addition and 2) edge deletion.

For edge addition, the trained link prediction module is utilized to compute the link prediction scores between each replayed node and all other nodes. We connect each replayed node with $N$ nodes based on the score, where $N$ is a hyperparameter, as follows:

$$\tilde{A}_{b_i j} = \begin{cases} 1, & \text{if } v_j \in \mathcal{K}_{b_i} \cup \mathcal{N}(v_{b_i}) \\ 0, & \text{otherwise} \end{cases} \quad (9)$$

where $\mathcal{K}_{b_i}$ denotes the set of $N$ nodes among all nodes in the graph, whose link prediction scores with a replayed node $v_{b_i}$ is the highest, i.e., $\mathcal{K}_{b_i} = \{\text{argmax}_{v_j}^{(N)} S_{b_i j}\}$. Note that although connecting nodes whose scores exceed a certain threshold is another option, we observed that this method leads to a significant number of edges being connected to a replayed node, which empirically results in an unsatisfactory performance.

For edge deletion, the candidate edges to be deleted are those that are originally connected to the replayed nodes. We remove the edges whose link prediction scores do not surpass a certain threshold. More formally, the structure information between a replayed node $v_{b_i}$ and its neighbor $v_j \in \mathcal{N}(v_{b_i})$ is updated as follows:

$$\tilde{A}_{b_i j} = \begin{cases} 1, & \text{if } S_{b_i j} > \tau \\ 0, & \text{otherwise} \end{cases}, \quad (10)$$

where $\tau$ is a hyperparameter indicating the threshold of the score.

Finally, through the above process of structure inference, we update the adjacency matrix $A^t$ for task $T_t$ to $\tilde{A}^t$.

#### 4.2.3 Discussion: Improving Scalability of Structure Inference.
Despite the effectiveness of the structure inference process described above, selecting $N$ nodes among all nodes in the graph as candidates to be connected to a replayed node requires the computation of $O(|V^t| \cdot |\mathcal{B}|)$, which is highly inefficient, where $V^t$ is the set of nodes in Task $T_t$. Therefore, to reduce the computational burden and make the model practical, for each replayed node, we only consider its top-$K$ closest nodes in the embedding space as the candidates to be connected to it, so that the computational burden can be reduced to $O(K \cdot |\mathcal{B}|)$, where $K \ll |V^t|$. Specifically, the candidate set $\mathcal{P}_{b_i}$ for a replayed node $v_{b_i}$ includes its top-$K$ closest nodes in the embedding space, which is formally defined as follows:

$$\mathcal{P}_{b_i} = \{v_{p_1}, \cdots, v_{pK}\} = \{\operatorname{argmin}_{v_{p_j}}^{(K)} dist(h_{b_i}, h_{p_j})\}, \tag{11}$$

where $h_{b_i}$ and $h_{p_j}$ are the embedding of node $v_{b_i}$ and node $v_{p_j}$ obtained from $\text{GNN}_{\theta^{t-1}}$ of task $T_{t-1}$, respectively, and $|\mathcal{P}_{b_i}| = K$. Then, we use the link prediction module $\text{LP}_{\phi^t}$ of task $T_t$ to find $N$ nodes with the highest link prediction scores among the nodes in the candidate set $\mathcal{P}_{b_i}$. Note that although we greatly reduce the number of candidate nodes to be connected to a replayed node, we expect the performance to be retained, since the replayed nodes are already selected in a manner that the coverage of each replayed node is maximized. This indicates that considering only a few nearby nodes as candidates to be connected allows the model to retain knowledge across the entire embedding space of all classes. In fact, we even expect that the above method of selecting a few candidates could outperform the case of using the entire set of nodes, since the candidates cannot contain nodes in the current task in which new classes are introduced[5]. This naturally prevents the replayed nodes from being connected to the current task's nodes, whose labels differ from the replayed nodes, which in turn allows the replayed nodes to maintain the homophily compared with using the entire set of nodes in every task. For detailed analysis, please refer to Section 5.2.

### 4.3 Downstream task: Node Classification
With the refined adjacency matrix $\tilde{A}^t$ derived from the graph structure learning process, we proceed with the downstream task, which is node classification. For task $T_t$, the training set consists of the training nodes from the current task, $\mathcal{D}_t{}^{tr}$, and the replay buffer, $\mathcal{B}$, stored until task $T_{t-1}$. The node classification loss is defined as follows:

$$\mathcal{L}_{cls} = \beta \mathcal{L}_{\mathcal{D}_t{}^{tr}}(\theta^t; \tilde{A}^t, X^t) + (1 - \beta)\mathcal{L}_{\mathcal{B}}(\theta^t; \tilde{A}^t, X^t), \tag{12}$$

where $\beta$ is a hyperparameter that balances the loss of the current task and the loss from the replay buffer, which is equivalent to $\beta$ in Equation 7, and $\mathcal{L}_{\mathcal{D}_i{}^{tr}}$ and $\mathcal{L}_{\mathcal{B}}$ denote the cross entropy loss for $\mathcal{D}_i{}^{tr}$ and $\mathcal{B}$, respectively. A large $\beta$ makes the model focus on the current task, while a small $\beta$ directs the model's attention toward the replay buffer to minimize catastrophic forgetting. The algorithm for a specific task of DSLR is summarized in Algorithm 1.

---

[5]This is because the candidates are selected when training the previous task $T_{t-1}$, and the structure inference is conducted in the current task $T_t$ whose classes do not overlap with those in $T_{t-1}$.

---

**Algorithm 1:** Framework of DSLR at task $T_t$

**Input:** Given task $T_t$, $\mathcal{G}^t = (A^t, X^t)$: Graph, $\text{GNN}_{\theta^t}$: GNN for node classification parameterized by $\theta^t$, $\text{LP}_{\phi^t}$: GNN for link prediction parameterized by $\phi^t$, $\mathcal{B}$: replay buffer.

**Output:** $\text{GNN}_{\theta^t}$ which can mitigate catastrophic forgetting of previous tasks, $\mathcal{B}$: updated replay buffer, $\mathcal{P}$: candidate set for each replayed node.

1 **if** $t \neq 1$ **then**
2      Initialize $\theta^t = \theta^{t-1}$, Randomly initialize $\phi^t$
     /* Train link prediction module              */
3      Evaluate $\mathcal{L}_{link}$ and $\mathcal{L}_{node}$      // Eq. 6 and 7
4      Evaluate $\mathcal{L}_{LP} = \lambda \mathcal{L}_{link} + (1 - \lambda)\mathcal{L}_{node}$      // Eq. 8
5      Update parameters : $\phi^t = \operatorname{argmin}_{\phi^t}(\mathcal{L}_{LP})$
     /* Structure inference                    */
6      **for** $v_{b_i}$ in $\mathcal{B}$ **do**
7          Add edges & Delete edges      // Eq. 9 and 10
8      **end**
9 **else**
10      Randomly initialize $\theta^t$
11 **end**
     /* Node classification (Downstream task)      */
12 Evaluate loss $\mathcal{L}_{cls}$      // Eq 12
13 Update parameters : $\theta^t = \operatorname{argmin}_{\theta^t}(\mathcal{L}_{cls})$
     /* Buffer selection                    */
14 **for** $C_l$ in $\mathbb{C}_{T_t}$ **do**
15      $\mathcal{B}_{C_l} \leftarrow$ Select $e_l$ replayed nodes for class $C_l$      // Eq. 4
16      $\mathcal{B} = \mathcal{B} \cup \mathcal{B}_{C_l}$
17 **end**
     /* Selecting candidates to be connected to replayed nodes      */
18 **for** $v_{b_i}$ in $\mathcal{B}$ **do**
19      Designate candidate set $\mathcal{P}_{b_i}$      // Eq. 11
20 **end**

## 5 EXPERIMENTS

***Datasets***. To evaluate DSLR, we use four datasets containing 3 citation networks, namely Cora [21], Citeseer [21], OGB-arxiv [9], and a co-purchase network, Amazon Computer [22]. The detailed description of the datasets is provided in Appendix C.

***Baselines***. We compare DSLR with recent state-of-the-art methods including rehearsal-based GCL methods. For more detail regarding the baselines, please refer to Appendix D.

***Evaluation protocol***. We use two metrics that are commonly used in continual learning research [15, 16, 20, 34]: 1) **PM** (Performance Mean) $= \frac{1}{T}\sum_{i=1}^{T} A_{T,i}$, and 2) **FM** (Forgetting Mean) $= \frac{1}{T-1}\sum_{i=1}^{T-1} A_{T,i} - A_{i,i}$. $T$ represents the total number of tasks, and $A_{i,j}$ denotes the accuracy of task $j$ after the completion of task $i$. PM is the average of the performance of each task after learning all the tasks[6], and FM is computed by taking average of decline in performances of a certain task[7]. For PM, higher values indicate better performance, while for FM, lower values indicate better performance. Please refer to Appendix E and F for more detail regarding experimental settings and implementaion details, respectively.

---

[6]Given three tasks, we predict task 1, task 2, and task 3 after the model learns all three tasks, and take the average of the three accuracies.
[7]For instance, forgetting of task 1 is the sum of the decrease in performance of task 1 after completing task 2 and task 3.

**Table 2: Model performance in terms of PM and FM. The buffer size is approximately 5% of the number of training nodes.**

| Datasets | Cora | | Citeseer | | Amazon Computer | | OGB-arxiv | |
|---|---|---|---|---|---|---|---|---|
| Metrics
Methods | PM ↑ | FM ↓ | PM ↑ | FM ↓ | PM ↑ | FM ↓ | PM ↑ | FM ↓ |
| LWF | 61.00 ± 4.47 | 25.73 ± 9.26 | 50.38 ± 2.02 | 21.37 ± 4.33 | 30.28 ± 1.11 | 80.71 ± 1.68 | 24.18 ± 2.69 | 48.56 ± 8.07 |
| EWC | 70.56 ± 3.13 | 31.90 ± 4.38 | 60.98 ± 3.45 | 21.56 ± 4.39 | 49.63 ± 4.27 | 49.62 ± 5.73 | 45.71 ± 6.50 | 30.91 ± 2.73 |
| GEM | 65.44 ± 5.16 | 32.97 ± 3.94 | 60.14 ± 1.72 | 21.89 ± 2.82 | 40.74 ± 3.03 | 42.19 ± 4.52 | 40.58 ± 4.26 | 29.28 ± 7.56 |
| MAS | 72.10 ± 5.25 | 17.21 ± 5.35 | 60.62 ± 3.32 | 23.44 ± 3.73 | 63.37 ± 1.80 | 23.17 ± 8.18 | 39.29 ± 2.91 | 30.36 ± 3.74 |
| ContinualGNN | 72.21 ± 1.83 | 33.84 ± 2.74 | 60.58 ± 0.86 | 34.89 ± 1.50 | 76.12 ± 0.75 | 29.33 ± 1.03 | 48.91 ± 4.15 | 52.83 ± 1.09 |
| TWP | 71.87 ± 8.45 | 25.77 ± 4.38 | 61.80 ± 1.31 | 24.76 ± 3.93 | 71.28 ± 3.26 | 26.55 ± 3.28 | 39.20 ± 5.92 | 25.65 ± 4.26 |
| ER-GNN | 78.68 ± 2.10 | 21.16 ± 3.52 | 65.49 ± 1.00 | 30.04 ± 1.19 | 77.20 ± 2.11 | 22.00 ± 2.13 | 37.19 ± 2.50 | 37.26 ± 1.55 |
| RCLG | 70.77 ± 4.74 | 15.71 ± 4.01 | 66.60 ± 3.33 | 22.67 ± 5.49 | 51.91 ± 6.57 | 16.71 ± 9.74 | 50.04 ± 6.44 | 41.00 ± 8.16 |
| **DSLR** | **81.59 ± 1.65** | **14.59 ± 2.61** | **69.54 ± 0.74** | **18.21 ± 0.96** | **80.08 ± 0.98** | **14.18 ± 3.15** | **51.46 ± 1.50** | **22.21 ± 3.82** |

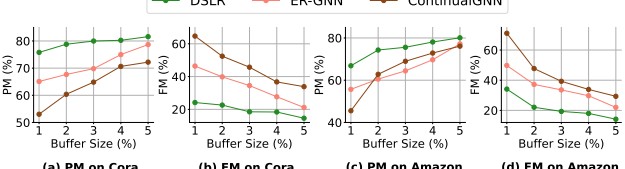

Figure 4: Performance of rehearsal-based approaches over various sizes of replay buffer.

Figure 5: Effect of considering diversity of replayed nodes.

## 5.1 Experimental Results

*5.1.1 **Overall Results**.* The experimental results on four datasets are summarized in Table 2. We make the following key observations: **(1)** In general, DSLR exhibits superior performance in terms of both PM and FM over all baselines, including recent GCL models utilizing rehearsal-based approaches. Furthermore, DSLR demonstrates relatively low variance across the 10 runs, indicating its ability to consistently perform well in various scenarios. **(2)** Recent rehearsal-based approaches such as ContinualGNN and ER-GNN outperform other baselines in terms of PM. However, in terms of FM, their performance is not consistently superior to other baselines, i.e., FM of ContinualGNN and ER-GNN is shown to be only slightly better or even worse compared with other baselines. This indicates that they fall short of retaining knowledge acquired in each task. This aligns with our expectation that the replayed nodes tend to cluster in certain regions of the embedding space of each class when the diversity aspect is overlooked, resulting in a lower performance on test nodes that fall outside of those regions. In contrast, DSLR addresses this issue by considering the diversity of the replayed nodes, leading to a superior FM. **(3)** We demonstrate the memory efficiency of DSLR. Fig. 4 illustrates the performance of different models that proposed their own replay buffer selection methods as the buffer size varies. Here, buffer size denotes the ratio of the number of replayed nodes to the total size of the train dataset. It is noteworthy that ContinualGNN and ER-GNN experience a significant decrease in PM and a sharp increase in FM as the buffer size decreases, while the performance change in DSLR is moderate. This indicates that DSLR can achieve comparable performance with a much smaller buffer size compared with other baselines, *demonstrating its practicality as it is crucial to use less memory in rehearsal-based GCL methods.*

*5.1.2 **Effectiveness of Coverage-based Diversity for Replay Buffer.*** Here, we conduct experiments to validate the effectiveness

of our proposed coverage-based diversity (CD) approach used to select nodes to be replayed (Fig. 5 and Table 3). We use two datasets from two distinct domains, i.e., Cora (citation network) and Amazon Computer (co-purchase network), as well as a large dataset, OGB-arxiv[8].

In Fig. 5, we observe that DSLR outperforms ER-GNN in both PM and FM regardless of the buffer size used. Note that for fair comparisons, we compare the performance of ER-GNN, which employs the mean feature (MF) approach, with DSLR, but without incorporating structure learning. This approach eliminates any advantages stemming from the structure learning component of DSLR. One notable point is that when the buffer size increases from 1% to 3% (i.e., small to mid-size), the increase in PM of DSLR is more significant than that of ER-GNN, and the decrease in FM of DSLR is also more highlighted compared to that of ER-GNN. This demonstrates that considering diversity through our proposed CD approach is more data-efficient than the MF approach adopted by ER-GNN. We conjecture that since ER-GNN selects nodes that are close to the average features of nodes in each class, similar nodes are continuously selected even when the buffer size increases. Conversely, as DSLR considers diversity through CD, a larger buffer size leads to the inclusion of more diverse and informative nodes by covering a wide representation space, thereby enriching the information contained within the replay buffer and effectively representing the whole class. Moreover, as the buffer size increases beyond 3%, the performance gap tends to decrease. This is because a larger buffer size diminishes the discriminative power of the replay buffer. However, as it is crucial to use a minimal buffer size in the rehearsal-based GCL, we argue that DSLR is practical in reality.

Next, in Table 3, we report the average diversity of the replayed nodes (i.e., Buff. Div.) and that of test nodes that are correctly predicted (i.e., Corr. Div.) over different classes. More precisely,

---

[8]Results on some datasets are presented in Appendix G.

**Table 3: Comparisons of diversity of replayed nodes ('Buff. Div.') and correctly predicted nodes ('Corr. Div.').**

| Datasets | Cora | | Amazon | | OGB-arxiv | |
|---|---|---|---|---|---|---|
| Method | Buff. Div. | Corr. Div. | Buff. Div. | Corr. Div. | Buff. Div. | Corr. Div. |
| ER-GNN | 0.55 | 0.62 | 0.59 | 0.58 | 0.77 | 0.74 |
| DSLR w/o SL | 1.46 | 0.97 | 1.31 | 0.82 | 1.58 | 1.09 |

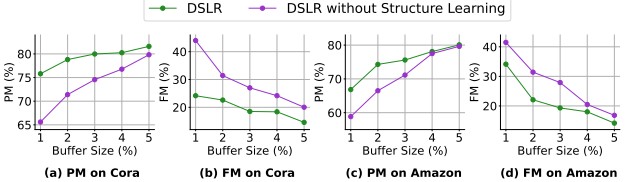

**Figure 6: Effect of structure learning for replayed nodes.**

'Buff. Div.' for class $C_l$ is calculated as $\frac{\mathbb{E}_{v_{b_m}, v_{b_n} \in \mathcal{B}_{C_l}} [dist(h_{b_m}, h_{b_n})]}{\mathbb{E}_{v_i, v_j \in train_{C_l}} [dist(h_i, h_j)]}$, i.e., the ratio of the sum of all pair distances between replayed nodes of class $C_l$ to that between training nodes of class $C_l$. In other words, a high 'Buff. Div.' implies that the distance between replayed nodes is relatively far, indicating that the buffer is diverse. Moreover, 'Corr. Div.' for class $C_l$ is calculated as $\frac{\mathbb{E}_{v_i \in corr_{C_l}} [dist(h_i, center_{C_l})]}{\mathbb{E}_{v_j \in test_{C_l}} [dist(h_j, center_{C_l})]}$, i.e., the ratio of the sum of distances between the correctly predicted nodes of class $C_l$ and the center of class $C_l$ to that between the test nodes of class $C_l$ and the center of class $C_l$. In other words, a high 'Corr. Div.' implies that the correctly predicted nodes are relatively evenly distributed from the class center, indicating that the model does not make predictions biased to certain regions. In Table 3, we observe that DSLR shows higher 'Buff. Div.' and 'Corr. Div.' compared with ER-GNN across all datasets. This aligns with our earlier discussion that the MF approach of ER-GNN tends to concentrate the replayed nodes around the class center, which in turn allows ER-GNN to only perform well on nodes located near the class center. In contrast, DSLR is capable of making accurate predictions regardless of the location of test nodes in the embedding space, which helps avoid overfitting to certain regions, thereby alleviating catastrophic forgetting. It is important to note that these results are obtained without employing structure learning, and thus highlight the effectiveness of our proposed CD approach.

5.1.3 **Effectiveness of Structure Learning.** Here, we demonstrate the effectiveness of structure learning in DSLR through various experiments (Fig. 6 and Fig. 7).

In Fig. 6, we observe that the structure learning component of DSLR consistently improves the performance of DSLR, and the performance gap between them becomes larger as the buffer size decreases. This indicates that our proposed structure learning component not only benefits the performance but also improves memory efficiency. Specifically, as shown in Fig. 6 (a), DSLR using only 1% of the buffer size performs on par with DSLR without structure learning that uses 4% of buffer size.

In Fig. 7, we report how PM and FM vary according to the degree of structure learning[9]. We observe that as structure learning is applied to a greater number of replayed nodes (i.e., increasing SL

---

[9]The label "SL ratio" represents the percentage of replayed nodes to which structure learning was applied, while "# top $N$" indicates that only the top N nodes with the highest scores among the candidates were connected to the replayed nodes.

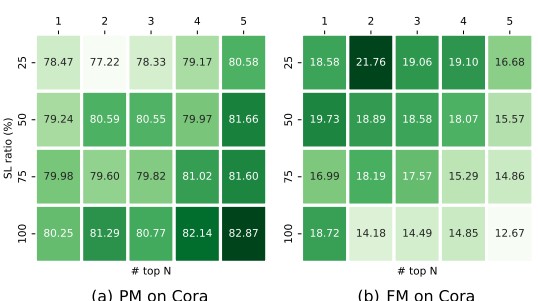

**Figure 7: Impact of structure learning for replayed nodes.**

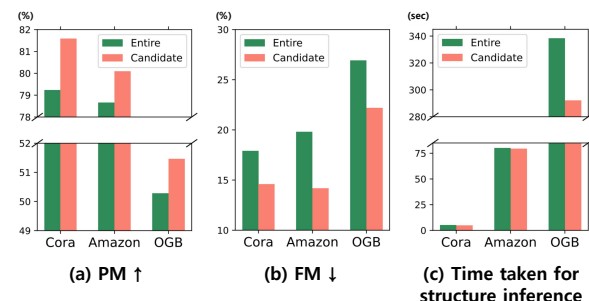

**Figure 8: Effectiveness and time efficiency of structure inference only for candidates.**

ratio), PM generally increases while FM decreases. This clearly demonstrates that applying structure learning to replayed nodes enhances the model performance. However, increasing $N$, that is, connecting the replayed nodes with more nodes does not always yield a better performance, implying that the value of $N$ should be carefully found.

## 5.2 Further Analysis

5.2.1 **Effectiveness of Selecting Candidates.** Recall that to improve the scalability of the structure inference stage described in Section 4.2.2, we do not consider the entire set of nodes as the candidates to be connected to the replayed nodes. Instead, we select a few nodes that are close to each replayed node in the embedding space as candidates for structure inference (as described in Section 4.2.3). Fig. 8 illustrates the PM, FM, and the total inference time (in seconds) during structure inference when considering the entire set of nodes (i.e., 'Entire') and when following our proposed candidate selection approach (i.e., 'Candidate'). In Fig. 8 (c), we observe that utilizing candidates for structure inference requires less time consumption compared with considering the entire set of nodes. Although the difference is subtle for relatively smaller datasets such as Cora and Amazon, notably lower inference time is observed for a large OGB-arxiv dataset when only considering candidates. This corroborates our argument that performing structure inference only for a few candidates is efficient. Moreover, in Fig. 8 (a) and (b), we observe that our proposed candidate selection method even outperforms the case of using the entire set of nodes as candidates. In summary, our proposed candidate selection method is both efficient and effective, demonstrating the practicality of DSLR.

5.2.2 **Ablation Study.** To comprehensively evaluate the impact of considering diversity when selecting nodes to be replayed, and

**Table 4: Ablation study on each component of DSLR.**

| Row | Component | | Cora | | Amazon | | OGB-arxiv | |
|-----|----|-----|-----|-----|-----|-----|-----|-----|
| | CD | Structure Learning | PM | FM | PM | FM | PM | FM |
| (1) | ✗ | ✗ | 78.68 | 21.16 | 77.20 | 22.00 | 37.19 | 37.26 |
| (2) | ✓ | ✗ | 79.82 | 20.03 | 79.63 | 16.82 | 48.48 | 26.15 |
| (3) | ✗ | ✓ | 80.42 | 18.14 | 78.82 | 18.25 | 46.38 | 28.09 |
| (4)-1 | ✓ | ✓ ($\mathcal{L}_{link}$) | 80.60 | 16.86 | 79.99 | 16.76 | 45.41 | 31.91 |
| (4)-2 | ✓ | ✓ ($\mathcal{L}_{node}$) | 79.92 | 18.85 | 77.03 | 20.36 | 47.84 | 26.5 |
| (4)-3 | ✓ | ✓ ($\mathcal{L}_{link}, \mathcal{L}_{node}$) | **81.59** | **14.59** | **80.08** | **14.18** | **51.46** | **22.21** |
| (5) | ✓ | Homophily ratio ↑ | 80.98 | 19.74 | 73.63 | 24.16 | 43.84 | 34.69 |

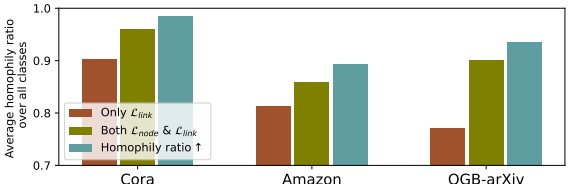

**Figure 9: Homophily ratio of replayed nodes under variants of structure learning.**

applying structure learning to the replayed nodes, we conduct an ablation study in Table 4. Note that Row (1) is equivalent to ER-GNN that adopts the MF approach for selecting replayed nodes, and Row (4)-3 is DSLR. We observe that adding either the CD component (Row (2)) or the structure learning component (Row (3)) is beneficial, while adding them both (Row (4)-3) performs the best. Moreover, to corroborate our argument that simply increasing the homophily ratio of the replay buffer is not effective (as shown in Fig. 2), and to advocate our proposed structure learning component, we deliberately added edges to replayed nodes in a way that the homophily ratio increases (Row (5)). We indeed observe that simply increasing the homophily ratio of the replay buffer performs poorly, especially in the large OGB-arxiv dataset. This indicates the importance of connecting replayed nodes to truly informative neighbors, which directly contributes to the performance of the continual learning.

Recall that $\mathcal{L}_{node}$ is included in the link prediction loss shown in Eq. 6 aiming at capturing the homophily aspect of the replayed nodes. In Fig. 9, we report the average homophily ratio of the replayed over all classes in three datasets. We observe that considering both $\mathcal{L}_{link}$ and $\mathcal{L}_{node}$ results in a higher homophily ratio compared with the case of using only $\mathcal{L}_{link}$ (i.e., $\lambda = 1$ in Eq. (8)), verifying that $\mathcal{L}_{node}$ helps increase the homophily ratio of the replayed nodes. Moreover, it is important to note that although simply increasing the homophily ratio (i.e., Homophily ratio ↑) yields the highest homophily ratio as expected, it results in a poor performance (See Row (5) in Table 4), verifying again that simply increasing the homophily ratio of the replay buffer is not an effective way to discovering truly informative neighbors.

*5.2.3* **Scalability Analysis**. We compare the average training time per task of DSLR with recent GCL models in Table 5. For the baseline models, we measure the time based on the provided official code. We observe that DSLR is highly efficient achieving up to 109 times faster training time compared with one of the state-of-the-art baselines, i.e., RCLG. Although ER-GNN is slightly more efficient than DSLR, considering the strong performance of DSLR from previous experiments, we argue that DSLR is a practical model that is both effective and efficient, which can also be efficiently trained on a large OGB-arxiv dataset.

**Table 5: Training time (in minutes) of recent GCL baselines.**

| Datasets | TWP | ContinualGNN | ER-GNN | RCLG | DSLR |
|----------|-----|--------------|--------|------|------|
| Cora | 0.16 (x1.79) | 0.33 (x3.69) | 0.08 (x0.85) | 9.90 (x109.99) | 0.09 (x1.00) |
| Amazon | 0.81 (x1.33) | 3.76 (x6.19) | 0.44 (x0.73) | 21.85 (x35.82) | 0.61 (x1.00) |
| OGB-arxiv | 2.40 (x1.35) | 18.56 (x10.42) | 1.26 (x0.71) | 49.91 (x28.04) | 1.78 (x1.00) |

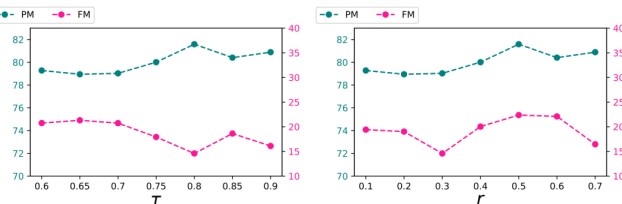

**Figure 10: Hyperparameter sensitivity analysis in Cora.**

*5.2.4* **Hyperparameter sensitivity analysis**. In this section, we provide a more in-depth analysis of the hyperparameters ($\tau$ and $r$) that significantly impact the performance of DSLR. More precisely, $\tau$ determines the threshold during structure inference, while $r$ determines the radius of coverage used in CD.

Fig. 10 illustrates how the model performance (i.e., PM, FM) changes while varying $\tau$ and $r$. The experiments are conducted for values around the optimal value that we identified. We observe that as for $\tau$, edges with less informative neighbors may not be adequately removed from the replayed node if it is too small. On the other hand, if $\tau$ is too large, edges with truly informative neighbors might get removed. This indicate that $\tau$ can have a negative impact on the model performance if it becomes too small or too large at the extremes. Once an appropriate $\tau$ is determined, stability is observed around its vicinity. Similarly, if $r$ is too small, even if there are many nodes of the same class around of a particular node, it might not be selected as a replayed node. Conversely, if $r$ is too large, nodes too far away could be included, undermining the consideration of diversity. Likewise, stability of the model performance is confirmed unless at extreme values. In short, DSLR outperforms baselines in most cases, once again demonstrating the superiority of our model.

We also present an analysis of another crucial hyperparameter $N$, which determines the number of neighbors connected through inference in Fig. 7 and Fig. 14 in Appendix.

## 6 CONCLUSION

In this paper, we propose a rehersal-based GCL model, called DSLR, that considers not only the class representativeness but also diversity within each class when selecting replayed nodes. We devise the coverage-based diversity (CD) approach to mitigate overfitting to specific regions in the embedding space, which helps avoid catastrophic forgetting. Additionally, we adopt graph structure learning to reformulate the graph structure in a way that allows the replayed nodes to be connected to truly informative neighbors, so that high-quality information can be propagated into the replayed nodes through message passing. DSLR demonstrates promising performance with a significantly smaller buffer size compared with baselines, demonstrating its practicality as it is crucial to use less memory in rehearsal-based GCL methods.

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

## A EXPERIMENTAL SETUP OF FIG. 2.

For simplicity of explanation, we only describe how the results for class $C_l$ is obtained. To perform evaluations on class $C_l$ for which $e_l$ replayed nodes need to be sampled, we first sort the nodes belonging to class $C_l$ based on their homophily ratio. Then, we sequentially split them into $\lceil |train_{C_l}|/e_l \rceil$ sets from the set with the smallest homophily ratio to the set with the largest homophily ratio. For example, if the total number of training nodes in class $C_1$ is 100 and $e_1$ is 20, the 100 nodes are sorted based on the homophily ratio, and they are split into 5 sets of 20 nodes each. For all other classes apart from class $C_l$, we randomly sample nodes to be replayed. That is, we use the $e_l$ nodes in the first set from class $C_l$, and the randomly sampled nodes from other classes as replayed nodes. Then, we report the forgetting performance, where the homophily ratio in $x$-axis is the average homophily ratio of the $e_l$ nodes in the first set from class $C_l$. We present the results in several intervals based on the average homophily ratio in each set, and report the averages and variances of forgetting based on the homophily ratio.

## B COMPLETE RELATED WORKS

### B.1 Graph Neural Networks

Various real-world data can be effectively represented in the form of graphs, encompassing domains such as social networks, molecules, and user-item interactions. As a result, research on graph neural networks (GNNs) is currently a topic of active study, which have found applications in various tasks, including node classification and link prediction. GNNs aggregate information from neighboring nodes, enabling them to capture both the structure and features of a graph, including more intricate graph patterns. One popular GNN model is Graph Convolutional Networks (GCN) [11], which introduces semi-supervised learning on graph-structured data using a convolutional neural network. GCN employs spectral graph convolution to update node representations based on their neighbors. However, GCN is a transductive model that relies on the adjacency matrix for training, requiring to retrain when the graph changes (e.g., with the addition of new nodes or edges) to perform tasks such as node classification. To address this limitation, GraphSAGE [8] is proposed as an inductive graph neural network model. Graph-SAGE trains an aggregator function that aggregates features from neighboring nodes, allowing it to handle unseen graphs. Another approach, Graph Attention Networks (GAT) [24], employs attention mechanisms to assign varying weights to neighbors based on their importance. GAT offers the advantage of efficiently extracting only the necessary information from neighbors, distinguishing it from other methods.

### B.2 Graph Structure Learning

Real-world graphs contain incomplete structure. To alleviate the effect of noise, recent studies have focused on enriching the structure of the graph. The objective of these studies is to mitigate the noise in the graph and improve the performance of graph representation learning by utilizing purified data. GAUG [33] leverages edge predictors to effectively encode class-homophilic structure, thereby enhancing intra-class edges and suppressing inter-class edges within a given graph structure. IDGL [4] jointly and iteratively learning the graph structure and embeddings optimized for enhancing the performance of the downstream task. SLAPS [7] utilizes self-supervision to learn structural information from unlabeled data by predicting missing edges in a graph, and NodeFormer [29] applies the transformer architecture for scalable structure learning. The application of structure learning in these methods improves the performance of downstream tasks by purifying the incomplete or noisy structure of the graph, taking into consideration the inherent characteristics of real-world graphs.

### B.3 Continual Learning

Continual learning, also known as lifelong learning, is a methodology where a model learns from a continuous stream of datasets while retaining knowledge from previous tasks. However, as the model progresses through tasks, it often experiences a decline in performance due to forgetting the knowledge acquired from past tasks. This phenomenon is referred to as catastrophic forgetting. The primary objective of continual learning is to minimize catastrophic forgetting, and there are three main approaches employed in continual learning methods: the rehearsal-based approach, architectural approach, and regularization-based approach.

**Rehearsal-based approach** aims to select and store important data that effectively represents the entire class from past tasks, using this data in subsequent tasks. This selected data is referred to as replayed nodes, and the set of the replayed nodes is called replay buffer. The primary objective of the rehearsal-based approach is to carefully select the optimal replayed nodes that prevent the model from forgetting knowledge acquired in previous tasks. Several strategies have been proposed for replay buffer selection. OCS [31] considered factors such as minibatch similarity, sample diversity, and coreset affinity in order to construct an effective replay buffer. Similarly, RAR [13] perturbed the replay buffer to make it similar to the current task data, thereby creating a robust decision boundary. Additionally, deep generative models have been utilized to generate the replay buffer, addressing memory constraints [23]. ER-GNN [34] proposes three buffer selection strategies, namely Mean of Feature, Coverage Maximization, and Influence Maximization. Note that the Coverage Maximization (CM) approach proposed in ER-GNN also aims to maximize the coverage of the embedding space of each class. Specifically, CM selects nodes in each class according to the number of nodes from other classes in the same task within a fixed distance. However, as the coverage for each node in CM is computed without considering the coverage of other nodes, we argue that the selected replayed nodes through CM are still at a risk of being concentrated in specific regions, which leads to a poor performance as shown in Table 6. On the other hand, since our proposed coverage-based diversity (CD) approach selects replayed nodes in a manner that maximizes the number of nodes included in the union of their coverage, the coverage of each node is computed considering the coverage of other nodes. Hence, unlike CM, CD always ensures that the selected replayed nodes are relatively more even spread in the embedding space.

**Table 6: Comparison of CM (ER-GNN) and CD (DSLR).**

| Datasets | Cora | | Amazon Computer | | OGB-arxiv | |
|---|---|---|---|---|---|---|
| Metrics Methods | PM ↑ | FM ↓ | PM ↑ | FM ↓ | PM ↑ | FM ↓ |
| CM (ER-GNN) | 78.13 ± 2.82 | 20.03 ± 3.95 | 78.93 ± 1.54 | 20.72 ± 1.88 | 46.84 ± 2.22 | 32.64 ± 3.47 |
| CD (DSLR) | 79.82 ± 2.70 | 20.03 ± 3.21 | 79.63 ± 2.49 | 16.82 ± 2.14 | 48.48 ± 3.25 | 26.15 ± 2.84 |

**Table 7: Statistics of evaluation datasets.**

| Dataset | # Nodes | # Edges | # Features | # Classes per task | # Tasks |
|---|---|---|---|---|---|
| Cora | 2,708 | 5,429 | 1,433 | 2 | 3 |
| Citeseer | 3,312 | 4,732 | 3,703 | 2 | 3 |
| Amazon Computer | 13,752 | 245,778 | 767 | 2 | 4 |
| OGB-arxiv | 169,343 | 1,166,243 | 128 | 3 | 5 |

**Architectural approach** involves modifying the model's architecture based on the task. If the model's capacity is deemed insufficient to effectively learn new knowledge, the architecture is expanded to accommodate the additional requirements. DEN [32] expands the capacity of the model selectively by duplicating components only when the loss exceeds a certain threshold. CPG [10] combines the architectural approach with the regularization approach. Similarly, CPG expands the model's weights if the accuracy target is not met. GCL [20] utilizes a reinforcement learning agent to guide the learning process and make decisions regarding the addition or deletion of hidden layer features within the network's architecture.

**Regularization-based approach** aims to regularize the model's parameters in order to minimize catastrophic forgetting while learning new tasks. This approach focuses on preserving important weights that were crucial for learning previous tasks, while allowing the remaining weights to adapt and learn new knowledge. One popular method in the regularization approach is EWC [12], which regularizes the changes in parameters that were learned in previous tasks. Another approach, Piggyback [18], achieves regularization by learning a binary mask that selectively affects the weights. By fixing the underlying network, Piggyback uses weight masking to enable the learning of multiple filters. Similarly, PackNet [19] utilizes binary masks to restrict changes in parameters that were deemed important in previous tasks, while allowing for flexibility in learning new tasks.

## C    DATASET DETAILS

We use four datasets to comprehensively evaluate the performance of DSLR, whose details are provided in this section. Detailed statistics of the datasets can be found in Table 7.

- **Cora** [21] is a static network that contains 2,708 documents, 5,429 links denoting the citations among the documents, and 1,433 features. The labels represent research fields. In our continual learning setting, the first 6 classes are selected and grouped into 3 tasks (2 classes per task).
- **Citeseer** [21] is a well known citation network containing 3,312 documents and 4,732 links and includes 6 classes. Similar to Cora, 6 classes are grouped into 3 tasks in our setting.
- **Amazon Computer** [22] is a co-purchase graph, where nodes represent products, edges denotes a co-purchase between two products, node features are bag-of-words encoded product reviews, and class labels indicate product category. In our setting, the dataset is divided into four tasks, excluding the two classes with the fewest number of nodes, resulting in 8 classes in total.
- **OGB-arxiv** [9] is a directed graph, representing the citation network between all Computer Science (CS) arXiv papers indexed by MAG [28]. Nodes denote arXiv papers, while directed edges denote citations from one paper to another. In our setting, to address the imbalance between classes, 15 largest classes

covering more than 80% of the entire dataset are selected and grouped into 5 tasks (3 classes per task).

## D    DETAILED DESCRIPTION OF BASELINES

- **LWF** [14] utilizes a combination of distillation and rehearsal techniques. The model distills the knowledge from the original model to a new model while training on the target task.
- **EWC** [12] uses a regularization term based on Fisher Information Matrtix that measures the sensitivity of the network's parameters to changes in the data. This term penalizes updates to the parameters that would cause significant changes in the parameters that were important for previous tasks.
- **GEM** [16] computes the gradients for the new task while storing the gradients of the previous tasks. It also uses memory projection to ensure that the gradients computed for the new task do not disrupt the performance on the previous tasks.
- **MAS** [1] updates the network's parameters while also updating and preserving the synaptic importance values. The model achieves this by utilizing a regularization term in the loss function, which penalizes large changes to the important synapses.
- **ContinualGNN** [26] incorporates new graph snapshots and updates the GNN accordingly, allowing the model to adapt to the evolving graph. It utilizes a graph distillation loss as a regularization term to retain learned knowledge from previous graphs while minimizing the impact of new graph updates.
- **TWP** [15] captures the topology information of graphs and detect the parameters that are crucial to the task/topology-related objective. It maintains the stability of crucial parameters aiming at preserving knowldge from previous tasks, while learning a new task.
- **ER-GNN** [34] is the state-of-the-art rehearsal based graph continual learning model. It samples the nodes that are the closest to the average feature vector, or the nodes that maximize the coverage of attribute/embedding space, or the nodes that have maximum influence.
- **RCLG**[10] [20] utilizes Reinforced Learning based Controller, which identifies the optimal numbers of hidden layer features to be added or deleted in the child network. The optimal actions are used to evolve the child network to train the model at each task.

We do not include SGNN-GR [27] as a baseline in our paper since the official source code is unavailable, which limits a fair comparison with other methods.

## E    DETAILED EXPERIMENTAL SETTING

In this paper, we follow a widely used experimental setting of GCL [15, 34] that transforms the benchmark dataset into an evolving graph, implementing practical continual learning in the real-world. For instance, when creating an evolving graph with three tasks from six classes of the Cora dataset, in the first task, only nodes from classes 1 and 2 are considered. Information about nodes from other classes or interclass edges connected to them are excluded, and only intraclass edges for class 1, intraclass edges for class 2, and interclass edges between classes 1 and 2 are used. After completing node classification for task 1, task 2 begins, introducing classes 3

---

[10]In the original paper, it was referred to as GCL, however, in this paper, to avoid confusion with Graph Continual Learning(GCL), it is named RCLG.

and 4. Again, nodes from classes 5 or 6 and any interclass edges associated with them are excluded. In node classification of task 2, we opt for a class-incremental setting where classes 1, 2, 3, and 4 are classified together, which is more practical and challenging compared to the task-incremental setting.

**Table 8: Hyperparameters of DSLR for each dataset.**

| Dataset | $\beta$ | $\lambda$ | $N$ | $K$ | $\tau$ | $r$ | Buffer size | Learning rate |
|---|---|---|---|---|---|---|---|---|
| Cora | 0.1 | 0.5 | 5 | 50 | 0.8 | 0.3 | 100 | 0.005 |
| Citeseer | 0.1 | 0.5 | 5 | 50 | 0.8 | 0.25 | 100 | 0.005 |
| Amazon Computer | 0.1 | 0.5 | 5 | 50 | 0.8 | 0.2 | 200 | 0.005 |
| OGB-arxiv | 0.05 | 0.5 | 5 | 50 | 0.8 | 0.15 | 3,000 | 0.005 |

## F IMPLEMENTATION DETAILS

Unless the replay buffer size is explicitly mentioned as in Figure 4, 5 and 6, the replay buffer size set to 100 for Cora and Citeseer, 200 for Amazon, 3,000 for OGB-arxiv dataset. This corresponds to approximately 5% of the trainset for each dataset. We report the average and standard deviation after running with 10 random seeds. DSLR utilizes several hyperparameters, i.e., $\beta$, $\lambda$, $N$, $K$, $\tau$, $r$, memory size for the replay buffer, and learning rate. We find the optimal hyperparameters for each baseline model through grid search. We tune them in certain ranges as follows: $\beta$ in {0.01, 0.05, 0.1, 0.2, 0.3}, $\lambda$ in {0, 0.25, 0.5, 0.75, 1}, $N$ in {1, 2, 3, 4, 5}, $K$ in {25, 50, 75, 100}, $\tau$ in {0.75, 0.8, 0.85, 0.9, 0.95}, $r$ in {0.1, 0.15, 0.2, 0.25, 0.3, 0.35, 0.4, 0.45, 0.5}, and learning rate in {0.0005, 0.001, 0.005, 0.01}. Table 8 shows specifications of detailed hyperparameters we used to present experimental result.

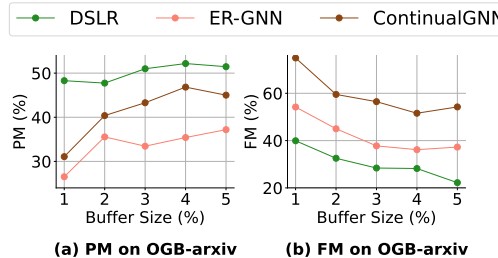

**Figure 11: Performance of rehearsal-based approaches over various replay buffer sizes.**

## G FURTHER ANALYSIS

### G.1 Comparison with rehearsal-based baselines on OGB-arxiv

We compare the performance of reheasal-based GCL baselines and DSLR on OGB-arxiv dataset in Fig. 11. Here, the buffer size refers to the proportion of the replayed nodes to the number of total train nodes. Similar to other datasets, we observe that DSLR outperforms other recent baselines in both PM and FM for all buffer sizes. Notably, as the buffer size decreases, the performance gap between DSLR and other baselines widens, indicating that DSLR is memory efficient, which is a primary goal of the rehearsal-based approach.

### G.2 Effect of considering diversity of replayed nodes on OGB-arxiv

We demonstrate the effectiveness of our proposed replay buffer selection method, i.e., coverage-based diversity (CD), on OGB-arxiv

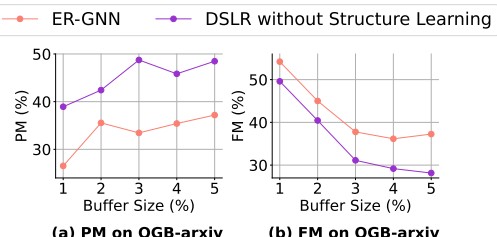

**Figure 12: Effect of considering diversity of replayed nodes.**

dataset. Fig. 12 illustrates the performance comparison between DSLR and ER-GNN, a state-of-the-art model that adopts the mean feature method for selecting nodes to be replayed. To clearly demonstrate the effect of considering the diversity of replayed nodes, we report the performance of DSLR *without structure learning*. Consistently across all buffer sizes, DSLR without structure learning outperforms ER-GNN in terms of PM and FM. This demonstrates that considering diversity proves to be more effective in preserving past knowledge with a small set of replayed nodes.

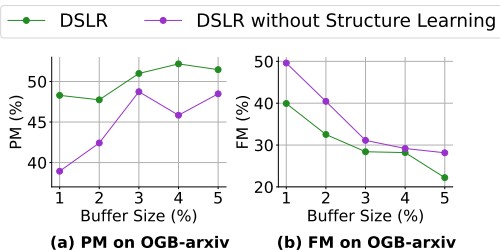

**Figure 13: Effect of structure learning for replayed nodes.**

### G.3 Effect of structure learning for replayed nodes on OGB-arxiv and Amazon

We demonstrate the effectiveness of our proposed graph structure learning. Fig. 13 compares the performance of the fully-fledged DSLR with that of DSLR *without structure learning*. We observe that DSLR with structure learning achieves similar performance compared with DSLR without structure learning even with a smaller replay buffer size. For instance, by examining Fig. 13 (a), it can be observed that when using structure learning, even with buffer sizes of 1% or 2%, DSLR can achieve a similar PM of DSLR *without structure learning* utilizing a buffer size of 3%. Similarly, Fig. 13 (b) demonstrates that the FM achieved by DSLR *without structure learning* using 2% buffer size can be attained with just 1% buffer size by employing structure learning. In other words, by connecting truly informative neighbors and ensuring that replayed nodes effectively represent the information of their respective classes, graph structure learning helps achieve a competitive performance with a much smaller replay buffer size. It is worth emphasizing once again that this highlights that DSLR is memory-efficient, which is a primary goal of the rehearsal-based approach.

To provide a clearer assessment of the effectiveness of structure learning, we further examine the performance variations based on the extent of structure learning applied. Fig. 14 illustrates the PM

**Table 9: Homophily ratio of the replay buffer with/without structure learning (SL) in DSLR.**

| Datasets | Class 1 | Class 2 | Class 3 | Class 4 | Class 5 | Class 6 | Class 7 | Class 8 | Class 9 | Class 10 | Class 11 | Class 12 |
|---|---|---|---|---|---|---|---|---|---|---|---|---|
| Cora without SL | 0.92 | 0.92 | 0.83 | 0.87 | - | - | - | - | - | - | - | - |
| Cora with SL | 0.97 (+0.05) | 0.97 (+0.05) | 0.95 (+0.12) | 0.95 (+0.08) | - | - | - | - | - | - | - | - |
| Amazon without SL | 0.80 | 0.84 | 0.95 | 0.68 | 0.83 | 0.59 | - | - | - | - | - | - |
| Amazon with SL | 0.84 (+0.04) | 0.98 (+0.14) | 0.97 (+0.02) | 0.75 (+0.07) | 0.86 (+0.03) | 0.63 (+0.04) | - | - | - | - | - | - |
| OGB-arxiv without SL | 0.98 | 0.97 | 0.94 | 0.83 | 0.84 | 0.98 | 0.81 | 0.88 | 0.67 | 0.87 | 0.63 | 0.96 |
| OGB-arxiv with SL | 0.97 (-0.01) | 0.97 (+0.00) | 0.93 (-0.01) | 0.93 (+0.10) | 0.90 (+0.06) | 0.99 (+0.01) | 0.86 (+0.05) | 0.88 (+0.00) | 0.72 (+0.05) | 0.91 (+0.04) | 0.64 (+0.01) | 0.97 (+0.01) |

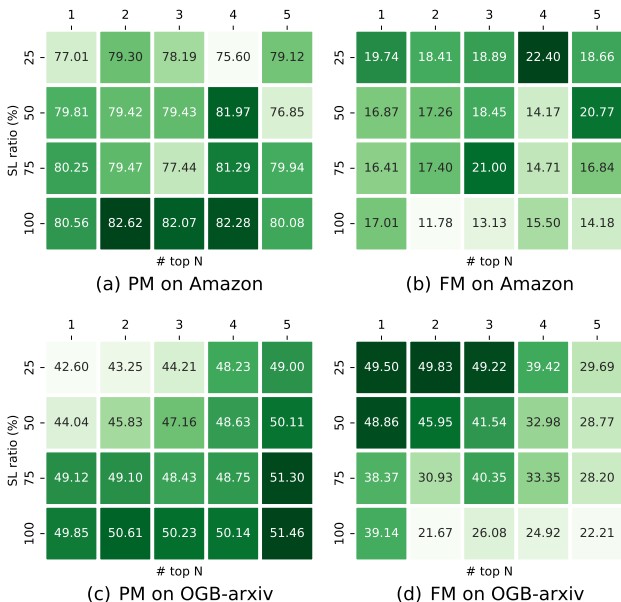

**Figure 14: Impact of structure learning for replayed nodes.**

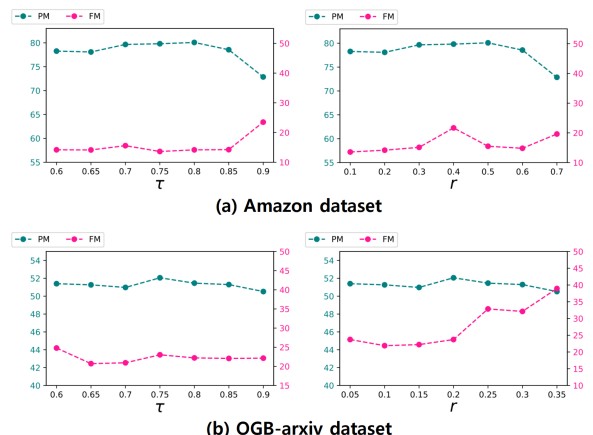

**Figure 15: Performance with varying hyperparameters.**

and FM metrics with respect to the degree of structure learning on OGB-arxiv and Amazon datasets. The "SL ratio (%)" represents the percentage of replayed nodes to which structure learning is applied, and "# top-$N$" indicates that the $N$ candidates with the highest scores are connected to the replayed nodes. We observe that applying the structure learning for more replayed nodes, meaning a higher "SL ratio," yields a clear improvements in terms of both PM and FM. On the other hand, when connecting more candidates to replayed nodes, i.e., increasing $N$, the effect on PM and FM is somewhat ambiguous. Nevertheless, the performance across various $N$ values is not sensitive, and we can observe that it consistently outperforms baselines. Finding the optimal $N$ based on data distribution could be a potential direction for future research.

Lastly, we verify that structure learning effectively increases the homophily ratio of nodes within each replay buffer, resulting in a positive impact on PM and FM. Table 9 presents the homophily ratio of replayed nodes for each class before and after applying structure learning on various datasets. As discussed earlier, we observe that the homophily ratio of replayed nodes increases after undergoing structure learning (See Table 9), resulting in an improved performance in both PM and FM metrics as shown in Fig. 6, and Fig. 13.

## G.4 Further Analysis for Hyperparameters on OGB-arxiv and Amazon

In Fig. 15, we provide a more in-depth analysis of the hyperparameters that significantly impact our model, namely $\tau$ and $r$. Specifically, $\tau$ determines the threshold during structure inference, while $r$ determines the radius of coverage. Fig. 15 illustrates how the model performance (i.e., PM, FM) changes with varying $\tau$ and $r$ values. The experiments are conducted for values around the optimal value that we found through grid search. If $\tau$ is too small, edges with less informative neighbors may not be adequately removed from the replayed node, while if $\tau$ is too large, edges with truly informative neighbors might get removed. This indicate that $\tau$ can have a negative impact on performance if it becomes too small or too large at the extremes. Once an appropriate $\tau$ is determined, stability is observed around its vicinity. Similarly, if the value of $r$ is too small, even if there are many nodes of the same class around of a particular node, it might not be selected as a replayed node. Conversely, if the value of $r$ is too large, nodes too far away could be included, undermining the consideration of diversity. With $r$ as well, stability of the model performance is confirmed unless at extreme values. In short, DSLR outperforms baselines in most cases, once again demonstrating its superiority. Note that the analysis of another crucial hyperparameter $N$, which determines the number of neighbors connected through inference, is presented in Fig. 7 and 14.

## H PSEUDOCODE

Algorithm 2 shows the pseudocode of CD.

---

**Algorithm 2:** Pseudocode of CD for class $C_l \in \mathbb{C}_{T_k}$

---

**Input:** Graph at task $T_t$ : $\mathcal{G}^t = (A^t, X^t)$, GNN for node classification parameterized by $\theta^t$, training set of class $C_l$ : $train_{C_l}$, empty replay buffer for class $C_l$ : $\mathcal{B}_{C_l}$.

**Output:** updated replay buffer : $\mathcal{B}_{C_l}$

1 Compute embedding of $train_{C_l}$ through $GNN_{\theta^t}$

   /* Compute the coverage of each node               */

2 **for** $v_i$ in $train_{C_l}$ **do**

3     $d = r \cdot E(v_i)$

4     $C(v_i) = \{v_j \mid dist(h_i, h_j) < d, \ y_i = y_j\}$

5 **end**

   /* Compute buffer size for class $C_l$              */

6 $e_l = \dfrac{\left|train_{C_l}\right|}{\sum_{t=1}^{t-1} \sum_{C_j \in \mathbb{C}_{T_t}} \left|train_{C_j}\right|} \cdot |\mathcal{B}|$

7 $count = 0$

8 $cover = \{\ \}$

9 $\mathcal{S} = train_{C_l}$

   /* Update $\mathcal{B}_{C_l}$ using CD approach             */

10 **while** $count \le e_l$ **do**

11     $v_{b_i} = \text{argmax}_{v_j \in \mathcal{S}}\left(\left|cover \cup C(v_j)\right|\right)$

12     $cover = cover \cup C(v_{b_i}) \cup v_{b_i}$

13     $\mathcal{S} = \mathcal{S} - C(v_{b_i})$

14     $\mathcal{B}_{C_l} = \mathcal{B}_{C_l} \cup v_{b_i}$

15     $count = count + 1$

16 **end**

---

Received 20 February 2007; revised 12 March 2009; accepted 5 June 2009

