# OpenReview forum: "DSLR: Diversity Enhancement and Structure Learning for Rehearsal-based Graph Continual Learning"
_ACM.org/TheWebConf/2024/Conference — TheWebConf24 Oral_

### Official Review · Reviewer_TY8g · 2023-11-07

**Novelty:** 5
**Technical Quality:** 5

**Review:**

This paper studies the rehearsal-based GCL, and proposes a CD selecting strategy and to improve the node quality via message passing.

1. The selection strategy is interesting. Why CD guarantee both representativeness and diversity. I think the two objectives are in contrast, how to balance them and achieve the trade-off.
2. To solve the drawback of MF, it should consider many other rehearsal methods, such as the simple randomness.
3. Typo error L369: Given the a graph

**Questions:**

see the review.

**Ethics Review Description:**

No needed.

**Reviewer Confidence:**

3: The reviewer is confident but not certain that the evaluation is correct

**Scope:**

4: The work is relevant to the Web and to the track, and is of broad interest to the community

---

### Official Review · Reviewer_qW6q · 2023-11-21

**Novelty:** 6
**Technical Quality:** 5

**Review:**

This paper investigates replay buffers in rehearsal-based approaches for graph continual learning (GCL). Existing methods tend to concentrate replayed nodes around class centers, risking overfitting and exacerbating catastrophic forgetting. To address this, we propose DSLR, a GCL model with two components: CD for selecting replayed nodes, considering class representativeness and diversity; GSL for enhancing graph structure, ensuring replayed nodes connect to truly informative neighbors. Extensive experiments demonstrate DSLR's superiority over state-of-the-art GCL methods, even with a small replay buffer size. Hence, if the authors revise their paper, I am pleased to give a positive judgment:

Overall, this paper exhibits sufficient innovativeness, but there are still some issues that need attention.

1. The segmentation of the two bars in Figure 8 may lead to misunderstandings. Consider revising the visualization to enhance clarity.
2. There is an abundance of descriptive text in the experiments, but specific data explanations are lacking. Please provide more detailed explanations of the data presented in the experiments.
3. It is recommended to include references for the models mentioned in the tables.
4. The last line in the appendix, "Received 20 February 2007; revised 12 March 2009; accepted 5 June 2009," seems misplaced or irrelevant. Please clarify or remove this line.
5. The number of references is relatively low, and most of them are from before 2022. It is recommended to consider adding more recent research papers to enhance the relevance and currency of the references.

**Questions:**

This paper investigates replay buffers in rehearsal-based approaches for graph continual learning (GCL). Existing methods tend to concentrate replayed nodes around class centers, risking overfitting and exacerbating catastrophic forgetting. To address this, we propose DSLR, a GCL model with two components: CD for selecting replayed nodes, considering class representativeness and diversity; GSL for enhancing graph structure, ensuring replayed nodes connect to truly informative neighbors. Extensive experiments demonstrate DSLR's superiority over state-of-the-art GCL methods, even with a small replay buffer size. Hence, if the authors revise their paper, I am pleased to give a positive judgment:

Overall, this paper exhibits sufficient innovativeness, but there are still some issues that need attention.

1. The segmentation of the two bars in Figure 8 may lead to misunderstandings. Consider revising the visualization to enhance clarity.
2. There is an abundance of descriptive text in the experiments, but specific data explanations are lacking. Please provide more detailed explanations of the data presented in the experiments.
3. It is recommended to include references for the models mentioned in the tables.
4. The last line in the appendix, "Received 20 February 2007; revised 12 March 2009; accepted 5 June 2009," seems misplaced or irrelevant. Please clarify or remove this line.
5. The number of references is relatively low, and most of them are from before 2022. It is recommended to consider adding more recent research papers to enhance the relevance and currency of the references.

**Ethics Review Description:**

-

**Reviewer Confidence:**

4: The reviewer is certain that the evaluation is correct and very familiar with the relevant literature

**Scope:**

4: The work is relevant to the Web and to the track, and is of broad interest to the community

---

### Official Review · Reviewer_rydM · 2023-11-22

**Novelty:** 6
**Technical Quality:** 5

**Review:**

This paper introduces a continual learning method for graph network. The authors start from the problem of concentrated replayed node and propose a new method to sample diverse nodes. Additionally, the diverse nodes are connected to informative neighbors with graph structure learning. The experiment and ablation study are solid to verify the hypothesis and method effectiveness.

Strength:
1) Paper is well-written. The motivation, methodology, and experiment are clear to read.
2) The experiment and analysis are exhaustive. They are solid and verify the hypothesis and method effectiveness.

Weakness:
1) The dataset and experimental setting are very different from the baseline ER-GNN, resulting the lack of validation.

**Questions:**

1) If the primary comparison is between the proposed method and ER-GNN, why not employing the same datasets?

2) For the only shared dataset Amazon Computer, the experimental setting is different. ER-GNN uses 10 classes as 5 tasks, while the proposed method only uses 8 classes as 4 tasks. The authors add a reason "excluding the two classes with the fewest number of nodes", but it is not convincing.

**Reviewer Confidence:**

4: The reviewer is certain that the evaluation is correct and very familiar with the relevant literature

**Scope:**

4: The work is relevant to the Web and to the track, and is of broad interest to the community

---

### Official Review · Reviewer_7zbE · 2023-11-23

**Novelty:** 6
**Technical Quality:** 5

**Review:**

The paper presents DSLR, a method for graph continual learning. In continual learning, one wants to learn from new data without retraining on the entire dataset. The main challenge is catastrophic forgetting, where older tasks see a decline in performance after the model is trained with newer data. The paper proposes a novel method based on the use of a replay buffer, that stores a small amount of data from previous tasks for later user when training subsequent tasks. The method builds on two main contributions: build the replay buffer so to both capture class representativeness as well as the diversity in each class, while previous approaches ignored diversity; change the structure (edges) of the graphs so that nodes in the replay buffer are connected to more informative nodes. The proposed method is assessed with a wide experimental evaluation on 3 datasets, assessing the overall performance compared to previous approaches, the contribution of the diversity component and of the structure altering component of the approach, as well as other aspects, including an ablation study.

The paper is well written overall, and the main ideas and contributions are clearly presented. While the ideas of using a reply buffer and of changing the structure of the network have been proposed and explored before, the use of a diverse set of reply nodes and the specific approach for altering the graph structure are original. The experimental evaluation shows that the proposed approach consistently improves over previous approaches, even if the improvement is often limited and within the variance in the estimates of the performance measures (in fact, the main gain is to provide a smaller variance in performance).

PROS
- The paper is well written overall, with nice introduction to the problem, and nice motivation and intuition for each step.
- The idea of using diversity to choose nodes for the reply buffer is really nice and intuitive
- The experimental evaluation covers various aspects/components of the proposed approach

CONS
- The abstract is not geared for a general audience, and it is not clear for nonexperts in continual learning and rehearsal-based approaches.
- The overall gain provided by the proposed approach is somehow limited, with the main positive aspect being a reduction in the variance of the performance
- The paper does not provide a link to the code and there is no mention of whether the code will be made available

**Questions:**

- Can you provide an alternative abstract that is more clear for a general audience?
- Can you provide a link to an anonymous repository with the code for the method and to reproduce the experiments?
- In section 4.2.3, what do you mean with the sentence “… requires the computation of O(|V^t|\cdot |B|)”?

**Reviewer Confidence:**

3: The reviewer is confident but not certain that the evaluation is correct

**Scope:**

4: The work is relevant to the Web and to the track, and is of broad interest to the community

---

### Official Review · Reviewer_j92f · 2023-11-26

**Novelty:** 4
**Technical Quality:** 5

**Review:**

The authors propose a new rehearsal-based method for graph continual learning, which considers both the class representativeness and the diversity within each class of the replayed nodes. Moreover, they adopt graph structure learning to reformulate the graph structure through a trained link prediction module. Experimental results demonstrate the effectiveness of the proposed method in improving performance.

The proposed method is well motivated and easy to follow. The idea of selecting representative and diverse nodes is intuitive and reasonable. Furthermore, the authors introduce a new structure learning method that effectively utilizes the replayed nodes. However, there are some concerns that need to be addressed.

**Questions:**

1. In Figure 4, it is observed that the improvement of PM for DSLR is limited as the buffer size increases, while ER-GNN and ContinualGNN show significant improvements. Does this mean that the proposed DSLR method only works well for relatively small buffer sizes? It is recommended to analyze this situation and provide an explanation.

2. According to the definition of Forgetting Mean, it should be a negative value since the performance on previous tasks typically decreases after training on a new task. However, the results shown in Table 2 do not align with this definition. Is this a mistake? Please clarify.

3. The hyperparameter N in Equation (9) is important for the proposed method. Although the authors provide some experimental results, it is still not clear how it affects the performance. For instance, the authors use a threshold \tau in Equation (10) to define the operation of edge deletion, but it could also be similarly defined by argmin_{v_j} S. Can the authors explain why they chose to use a threshold \tau in Equation (10)?

4. It is suggested to provide more details on how D_link is constructed in Eq(6).

5. Since the proposed method involves buffer selection and training a link prediction module, it is suggested to discuss its efficiency compared to other methods.

6. The authors claim that a large \beta in Eq(12) makes the model focus on the current task, while a small \beta directs the model鈥檚 attention toward the replay buffer to minimize catastrophic forgetting. Can the authors provide the corresponding experimental evidence to support this claim? Furthermore, can you investigate the performance when cross-entropy is directly computed on both D_i^tr and B without introducing \beta in Equation (12)?

**Reviewer Confidence:**

3: The reviewer is confident but not certain that the evaluation is correct

**Scope:**

3: The work is somewhat relevant to the Web and to the track, and is of narrow interest to a sub-community

---

### Decision · Program_Chairs · 2024-01-22

**Decision:**

Accept (Oral)

**Comment:**

The paper proposes a new technique for graph continual learning, focusing on better ways of selecting the replay buffer (i.e. a small amount of data used for future training tasks to avoid catastrophic forgetting). It uses both the representativeness and the diversity within each class of replayed nodes, as well as graph structure learning to ensure replayed nodes are well-connected to informative neighbors. Experiments demonstrate the effectiveness and efficiency of the approach.

 The paper is clear, with interesting techniques and conclusive experiments. Although reviewer consensus is that the paper has broad appeal for TheWebConf, the area of graph continual learning is somewhat more niche.

 Strengths
 * Well-motivated and easy to follow.
 * The coverage-based diversity approach (using both representativeness and diversity) is intuitively well-justified.
 * Experiments are (after rebuttals) thorough, showing conclusive improvements.

 Weaknesses:
 * Recommendation to add more recent related work.